# Genetic and Genomic Analysis of Cow Mortality in the Israeli Holstein Population

**DOI:** 10.3390/genes14030588

**Published:** 2023-02-25

**Authors:** Joel Ira Weller, Ephraim Ezra, Eyal Seroussi, Moran Gershoni

**Affiliations:** 1Israel Cattle Breeders Association, Caesarea 38900, Israel; 2ARO, The Volcani Center, Rishon LeZion 15159, Israel

**Keywords:** genomics, cow mortality, genetic analysis, Israeli Holsteins

## Abstract

“Livability” was defined as the inverse of the probability of death. The objectives of this study were to estimate the heritability, genetic and phenotypic trends for the livability of Israeli Holstein cows; estimate the genetic and environmental correlations between livability and the nine traits included in the Israeli breeding index; estimate the effect of the inclusion of livability in the Israeli breeding index on expected genetic gains; and compute a genome-wide association study (GWAS) for livability. Seven data sets were analyzed. All data were derived from the database of the Israeli dairy cattle herd-book. The mean livability for the complete data set of 523,954 cows born from 2000 through 2016 was 89.6%. Pregnancy reduced livability by 15%. Livability generally increased with parity and days in milk within parity. Heritability of livability was 0.0082. Phenotypic and genetic trends over the 14-year period from 2000 through 2013 were −0.42% and −0.22% per year. If livability is included in the Israeli breeding index, accounting for 9% of the index, livability would increase by 1.3% and protein production would decrease by 11 kg over the next decade, as compared to the current index. A marker in proximity to the oxytocin–vasopressin locus had the greatest effect in the GWAS. Oxytocin activity in cattle affects calving-associated pathologies and maternal death. Inclusion of livability in the Israeli breeding index is not recommended.

## 1. Introduction

Dairy cow survival, or herd life, is an economically important trait that is now included in the selection indexes of most commercial dairy-cow populations [1]. The main reasons for cow culling are low production, failure to conceive and illness [2]. There is currently no consensus for the trait definition and methodology for evaluation across countries [2]. Some countries compute “functional herd life,” defined as herd life corrected for milk production [3]. In the US, days in milk >305 are not included in the calculation of herd life [4]. In Israel, herd life is computed simply as the number of days from first calving to culling [5].

Mortality rates (MORT) in the US Holstein population increased from 1998 through 2007 [6,7]. Annual MORT of 5.7% was reported for the US dairy industry in 2008 [7]. In 2016, for all US breeds, on average, 16.7% of cows died instead of being sold. Death losses averages 6% per lactation, but these were higher for later parities [8]. Although cows that exit the herd due to death are included in all common algorithms used to compute herd life, the loss is greater, because the farmer loses income from sale of the culled cow and must dispose of the carcass. VanRaden et al. [8], in 2016, estimated the mean loss as $1200/cow in the US. McConnel et al. [9] found that the main herd management factors associated with MORT in the US Holstein population were herd levels of respiratory disease, lameness, antibiotic use for treating sick cows, the percentage of culled cows less than 50 days in milk, the average calving interval, the use of a total mixed ration and the region of the country.

Owing to the significant additional loss that results from cow death, as compared to culling, “cow livability (LIV),” the inverse of death, has been included in the major US selection indexes since 2016 [8], even though all estimates of heritability for this trait are very low [6,7,8,10,11]. Similarly to herd life, genetic evaluation for LIV has the disadvantage that in most cases, records are not available when breeding decisions are made. Prior to genomics, the major decisions on returning bulls to general service were made based on first parity production records. In the last decade, most bulls in commercial dairy-cow populations in the US and Europe are used for general service at the age of one year, based on genomic evaluations [12]. However, accuracies of genomic evaluations of young bulls without daughter records decrease as the time difference between the computation of genomic evaluations and generation of daughter records increases. Therefore, predicted herd-life records are generally computed for cows and weighted accordingly in genetic evaluations. Predicted herd-life records for live cows are computed based on the length of the current herd life and other factors that affect herd life, such as lactation number, pregnancy status and current milk production [5]. At present, only a few studies have considered factors that affect deaths of dairy cows [6,7,10], and no studies have attempted to predict the probability of cow death based on these factors.

Computation of a genome-wide association study (GWAS) in dairy cattle requires computation of genetic evaluations for animals with genotypes. To compute unbiased genetic evaluations, it is first necessary to correct for environmental factors that may affect the phenotypic values of the traits analyzed. The objectives of this study were therefore: (1) to determine environmental factors affecting the LIV of Israeli Holsteins, (2) to estimate the heritability and herd-year-season’s effects on LIV by a restricted maximum likelihood (REML) analysis based on the individual animal model (IAM), (3) to compute genetic evaluations for LIV for all Israeli Holstein cows with valid records since 2000, (4) to estimate genetic and phenotypic trends for LIV, (5) to estimate the genetic and residual correlations between LIV and the nine traits included in the Israeli breeding index (PD20), (6) to estimate the effect of inclusion of LIV in the Israeli breeding index on expected genetic gains and (7) to compute a GWAS for LIV based on Israeli Holstein bulls with genotypes and genetic evaluations.

## 2. Materials and Methods

As in [8], in all analyses, LIV was defined so that cows with the exit reason of death were given a score of zero; all other cows were given a score of 100. Seven data sets were analyzed. Basic statistics of the data sets are given in Table 1. All data were derived from the database of the Israeli dairy cattle herd-book (http://www.icba-israel.com/, accessed on 2 January 2023). The first data set included valid exit records of 523,954 cows born from 2000 through 2016, with up to six parities, and no more than 30 months in milk at the last parity. This data set was used to estimate the effects of parity, exit month, days in milk at exit date, pregnancy status and milk production in the final lactation. The second data set included 475,512 cow records from data set 1 with valid first parity records for the 9 traits included in the Israeli breeding index. This data set was used to compute phenotypic correlations between LIV and the traits included in the current Israeli breeding index. The third data set included 174,106 cow records selected from data set 1 born from 2004 through 2009 with known sires of the Holstein breed. These data were used to compute REML variance components for LIV based on the “individual animal model” (IAM), and also included parents and grandparents of cows with LIV records. The fourth data set included 145,023 cows born from 2003 through 2009 with known Holstein sires, valid records for LIV and first parity records for 8 of the 9 traits included in PD20. For herd life, only a single record is generated per cow. The index traits are listed in Table 2. This data set was used to compute REML variance and covariance components among the 10 traits. As for data set 4 (Table A1, Table A2 and Table A3), this data set also included parents and grandparents of cows with records. The fifth data set included 511,438 cow records selected from data set 1 with known sires, but also included cows with sires of breeds other than Holstein (~1% of the total number of cows). This data set was used to compute genetic evaluations for LIV by the IAM. This analysis also included parents and grandparents of cows with records. The sixth data set included 1007 bulls with genetic evaluations for LIV with reliabilities >0.5. This data set was used to compute Pearson correlations between the genetic evaluations for LIV and the nine selection index traits. The final data set included 965 bulls of data set 6 with genomic evaluations. This data set was used for the GWAS analysis of LIV. 

Data set 1 was analyzed by PROC GLM of SAS [13] (Cary, NC, USA) using the following linear model:(1)$${\mathrm{Liv}}_{\mathrm{ijklmn}}\;=\;{\mathrm{nm}}_{i}\;+\;\mathrm{milk}\;+{\;\mathrm{milk}}^{2}+\;{\mathrm{preg}}_{j}\;+\;{\mathrm{lac}}_{k}\;+\;{\mathrm{mon}}_{l}\;+\;\mathrm{HYSm}\;+\;\dim\;+\;\mathrm{idim}\;+\;\mathrm{preg}\times \mathrm{lac}\;+\;\dim\times \mathrm{lac}\;+\;\mathrm{idim}\times \mathrm{lac}\;+\;e_{\mathrm{ijklmn}}$$
where: Liv_ijklmn_ is the LIV record for cow n with milking status i, pregnancy status j, lactation number k, exit month l and herd-year-season m; nm_i_ is the milking status—either a valid record was recorded for milk production in the last parity or not; milk and milk^2^ are linear and quadratic effects of predicted 305-day milk production in the last lactation for cows with at least 35 days in milk; preg_j_ is pregnancy status j on the date of cow exit, either pregnant or open; lac_k_ is parity number k at exit; mon_l_ is calendar month l of exit; HYS_m_ is herd-year-season m; dim is days in milk at the exit date; idim is the inverse of dim; preg×lac, dim×lac and idim×lac are the corresponding interactions; and e_ijklmn_ is the random residual. Since the last lactation was generally incomplete, predicted milk production to 305 days was computed as described previously [14]. HYS was defined based on freshening date for the cow’s last lactation. Two seasons were defined for each herd-year beginning in October and April. Separate HYS effects were defined for cows culled during first and later parities. Effects and significance were computed for each effect and the total model coefficient of determination.

As noted previously, 8 of the 9 traits included in PD20 were recorded on first-parity cows. Herd life was computed as the number of days from first calving to exit from the herd. Pearson correlations were computed among LIV and the 9 index traits for all cows included in data set 2.

Data set 3 was analyzed by the AIREMLf90 program [15] using the following IAM:(2)$${\mathrm{Liv}}_{\mathrm{ijklmn}}\;=\;a_{i}\;+\;{\mathrm{preg}}_{j}\;+\;{\mathrm{lac}}_{k}\;+\;{\mathrm{mon}}_{l}\;+\;{\mathrm{HYS}}_{m}\;+\;e_{\mathrm{ijklmn}}$$
where: a_i_ is the additive genetic effect of animal i, and the other terms are as defined previously. The additive genetic, HYS and residual effects were considered random, and the other effects were considered fixed. In addition to cows with records, additive genetic effects were computed for all known parents and grandparents of cows with records. Two fixed groups were defined for parents of animals with unknown parents: one for males and one for females. The numbers of animals with records, ancestors and HYS are given in Table 1. 

Data set 4 was analyzed by the MTC REML program [16], which can be applied to large data sets with multiple traits but does not estimate standard errors. The analysis model for all traits included the additive genetic animal effect and the HYS effect, as described previously, except that the year-season was set relative to the cow’s first parity freshening date, and cows culled in first and later parity were assigned to the same year-seasons. The HYS effect was assumed to be fixed. As for the analysis of data set 3, 2 genetic groups were defined, based on the sex of the animal with unknown parents. 

The vector of expected genetic changes over 10 yr (**Φ**) based on the genetic and residual variance components was computed using the following equation [17]:(3)$$\Phi =\;ibG/\left(b^{\prime }Pb\right)0.5$$
where i = the selection intensity; **b** = the vector of breeding index coefficients listed in Table 2; **G** = the genetic variance matrix; and **P** = the phenotypic variance matrix, computed as the sum of the genetic and residual variance matrices. As noted previously, VanRaden et al. [8] estimated the economic value of cow death as $1200/per cow, or $12 for 1% change in LIV. (Based on a rough calculation, the value is probably slightly higher under Israeli conditions.) Under the assumption that the value of 1 unit of PD20 = $0.25 [18], the index coefficient for LIV should be ~48. The selection intensity was set to 4, which roughly corresponds to the cumulative selection intensity over the 4 paths of selection obtained after 10 years in an advanced breeding program [17]. 

The contribution of each trait to the total selection index (*c_j_*) with and without inclusion of calf survival was computed as:(4)$$c_{j}=\frac{abs(b_{j}g_{j})}{\sum_{j=1}^{J}abs(b_{j}g_{j})}$$
where *b_j_* = the index coefficient for trait *j*; *g_j_* = the genetic standard deviation for trait *j*; *J* = total number of traits; and *abs* = absolute value. In addition, Equations (3) were also computed with a coefficient of 5 for LIV. In this case, the contribution of LIV would be 1% of the total index, as proposed by [8]. 

The analysis model for data set 5 included only the animal additive genetic effect, the HYS effect and random residual. In this model, the HYS effect was fixed, and the other two effects were random. As in data set 1, HYS was determined relative to the cow’s last freshening date. The dependent variables for this analysis were the residuals from a GLM analysis of LIV, including the preg, lac and mon effects described previously. A total of 64 groups for animals with unknown parents were defined based on the sex of each animal with unknown parents, its birth year, which parents were unknown and the breed of the sire. Approximate reliabilities were computed for all animals by the algorithm of Misztal and Wiggans [19]. Phenotypic and genetic trends were computed for cows born between 2000 and 2013. The phenotypic trend was computed as the regression of the LIV score from data set 1 based on the cows’ birth dates. Genetic trends were computed as the regression of the cows’ breeding values for LIV based on the cows’ birth dates. Cows with birthdates past 2013 were not included because of potential bias. As noted previously, cows still alive at the time of analysis were not included in the analysis. Since a significant fraction of cows born after 2013 are still alive, cows with exit records in the later years would be a biased sample with respect to death rates.

Holstein bulls with reliabilities >0.5 from the analysis of data set 5 were included in data set 6. This data set was used to compute Pearson correlations between the breeding values for LIV and the 9 traits included in PD20. Genetic evaluations for the 9 index traits were computed as described previously [5,14,20,21].

The GWAS analysis included 965 Holstein bulls from data set 6 with genotypes and phenotypes, born in or after 1991. All genotyping was performed by Neogen (Lansing, MI, USA, https://www.neogen.com/about, accessed on 2 January 2023) and BeadChips (Illumina Inc., San Diego, CA, USA, https://www.illumina.com/science/technology.html, accessed on 2 January 2023) on hair or semen samples collected in Israel by SION, the Israeli AI institute (http://www.sion-israel.com/english/, accessed on 2 January 2023). A bull’s genotype was considered valid if at least 85% of the SNPs had valid genotype calls. In addition, for males, >90% homozygosity for markers on the X chromosome was required. As genotyping of these bulls was performed using several SNP chip platforms with different qualities and coverage, we performed imputation using 2114 Holstein bulls with genomic evaluations. A total of 50,392 genetic markers were selected for further analysis, according to [22]. Following additional formatting to fit the requirements of the imputation software, the imputation process was carried out using the findhap.f90 v3.0 program [23] (USDA-ARS, Beltsville, MD, USA, https://aipl.arsusda.gov/software/findhap/, accessed on 2 January 2023).

The GWAS files were prepared and formatted as described in [18,24] using the Plink software [25] (https://www.cog-genomics.org/plink/, accessed on 2 January 2023). The response variable was the sires’ transmitting abilities (half of the estimated breeding value) for LIV. The allele substitution effects and the nominal probabilities for the hypothesis of no effect were computed using EMMAX software, version EMMAX-intel64-20120205 [26] (http://csg.sph.umich.edu/kang/emmax/, accessed on 2 January 2023). We generated a relationship matrix based on the identity by state matrix calculated using the emmax-kin-intel64 algorithm, using the -v -s -d 10 flags. Then, using the -v -d 10 –t flags, the relationship matrix and the –k argument, GWAS was computed. The nominal probability values were corrected for multiple testing by computation of the false discovery rate, as proposed by Benjamini and Hochberg [27]. The genome-wide significance threshold was set to *p* < 0.05. Using the relatedness matrix, we also assessed the variance explained by all SNPs by the EMMAX algorithm for genomic REML. 

QTL annotation was performed as follows: first, the annotation file containing the quantitative trait locus (QTL) database for cattle from the Animal QTLdb (https://www.animalgenome.org/cgi-bin/QTLdb/BT/index, accessed on 2 January 2023) and the gtf file with the annotated bovine genome from Ensembl (https://www.ensembl.org/Bos_taurus/Info/Index?db=core, accessed on 2 January 2023). Both files are based on the bovine reference genome assembly ARS-UCD1.2 [28], corresponding to this study’s markers coordinates. The GALLO package [29] (https://cran.r-project.org/web/packages/GALLO/index.html, accessed on 2 January 2023) was applied to annotate the SNP markers found for LIV and the genes spanning them. Gene enrichment analysis was performed using the Enrichr server [30] (https://maayanlab.cloud/Enrichr/, accessed on 2 January 2023), which can identify gene enrichment for several terms and data sources, including diseases, pathways, Gene Ontology (GO) terms and tissue expression. 

## 3. Results

### 3.1. Phenotypic Analysis of Livability

The mean LIV score for data set 1 was 89.6%—that is, a death rate of 10.4%. This value is higher than the value from the US dairy population of 5.7% from 2007 [7] but lower than the more recent value of 16.7% [8]. 

All effects included in the GLM analysis of model 1 were significant at *p* < 0.0001. The coefficient of determination was 0.29; thus, these factors explained nearly 30% of the phenotypic variation in LIV. Mean LIV scores by pregnancy status and last parity are given in Table 3. Nearly 90% of the cows at exit were open. Mean LIV scores for pregnant and open cows were 76.3% and 91.1%—that is, pregnant cows had a 15% higher death rate. Death rates were lowest for second parity, and highest for sixth. Two previous studies [6,7] found that death rates increased with parity, including the first. Mean LIV scores by exit month are given in Figure 1. The model effects from the analyses of data sets 1 and 3 are also given. Like previous results [6,7], all three analyses demonstrated that death rates were highest in the summer. In the current study, death rates were lowest in February, whereas Miller et al. [7] found the lowest death rate in November. 

Mean LIV as a function of predicted 305-day milk production in data set 1 is given in Figure 2. Mean LIV was only 75% for cows without lactation records for the current parity and was highest at 95% for cows with 305-day milk production between 7 and 9 tons. For cows with production of 14 tons, mean LIV decreased to <90. Miller et al. [7] found a similar relationship between milk production and death rates in the US populations for both Holsteins and Jerseys.

Mean LIV in data set 1 as a function of months in milk is plotted in Figure 3. For cows with <1 month in milk, mean LIV was 66% and increased to 90% for cows with >6 months in milk, and reached 98% after 21 months in milk. Although 10 months is generally considered the optimal lactation length, 10% of the cows had >18 months in milk. Miller et al. [7] found similar results with respect to the relationship between days in milk and death rate, but did not differentiate between cows with >251 days in milk.

The effect of milking vs. non-milking status at exit was 24.1%. That is, milking cows had 24% higher LIV than cows without a valid lactation record for the final parity. The linear and quadratic effects of 305-day milk production were −1.1% LIV per ton and −0.06 ton^2^. The linear and inverse effects of days in milk were 0.044 and −53.76. The effect of pregnancy status was 18.8. That is, pregnancy increased the probability of death by nearly 20%. This value is slightly larger than the difference in the simple means and the model effects from the analyses of data sets 3 and 5 listed in Table 2.

The effects of pregnancy status and exit from the analyses of data sets 3 and 5 are also listed in Table 3. The effects were similar to the differences in the simple means, even though the effects in data set 3 were derived by a REML analysis, and the effects in data set 5 were derived by a fixed linear model. LIV was highest in the second parity at 91.5% and declined by 5% for parity 6. The effects for pregnancy in both models were slightly less than the differences in the simple means.

### 3.2. Genetic Analysis of Livability

Mean LIV scores and breeding values of the cows in data set 5 by birth year are given in Figure 4. The genetic base was set so that the mean of cows born in 2015 would equal zero. Mean LIV declined by 6% for cows born in 2016 compared to cows born in 2000; and mean breeding values declined by 2.5% for cows born in 2011 compared to cows born in 2001. Since 2012, there has been a slight increase in mean breeding values. This may be due to the fact that only cows with exit dates were included in the analysis, and the cows born in the last few years may be a biased sample. Phenotypic and genetic regressions for LIV over the 14-year period from 2000 to 2013 were −0.42 and −0.22 per year. Subtracting the genetic from the phenotypic regression gives an environmental regression of −0.20%/year.

The variance components from the analysis of data set 3 were 6.8, 31.3 and 787.3 for the additive genetic, HYS and residual variance components. Thus, the HYS variance was <4% of the total variance. Heritability, computed as the additive genetic variance divided by the sum of the additive genetic and residual variance was 0.0082 with a standard error of 0.0016, as compared to the value of 0.012 from the multi-trait analysis of data set 4. These values correspond to most linear model values in the literature [7,8], although threshold estimates, which compute variance on an underlying continuous scale, are somewhat higher, as expected [6,10]. Miller et al. [7] also used the AIREMLF90 program and found a heritability of 0.013. VanRaden et al. [8] found heritabilities of 0.0043, 0.007, 0.0073, 0.0074 and 0.0074 for the first five parities, which were all equivalent to about 0.03 on the underlying continuous scale. Heritability estimates by a Gibbs sampling algorithm analysis of a threshold model ranged from 0.04 to 0.07 for separate analyses by parity and region for US Holsteins [6]. Tsuruta et al. [10] found heritabilities of 0.03 to 0.04 by parity with a bivariate threshold model, including MORT and 305-day milk production. 

The genetic and residual variance matrices from the REML analysis of data set 4, and the heritabilities and genetic and residual correlations are given in Appendix A. The genetic and environmental correlations of the 9 index traits with LIV are given in Table 4. The phenotypic correlations between LIV and first parity records for the nine index traits from data set 2 are also given in Table 4. The absolute values of all environmental correlations with LIV in data set 2 were < 0.1. In the REML analysis of data set 4, the environmental correlation with the largest absolute value was −0.06 with milk production. The genetic correlations with the milk production traits were all negative, with absolute values > 0.3. The genetic correlation with protein production was −0.42, which is the main trait in the Israeli breeding index (Table 2). Genetic correlations between MORT and 305-d milk yield for the first three parities were estimated as −0.01, 0.01 and 0.31 in the Southwest US, which is the region in the US that most closely approximates Israeli climatic conditions [6]. Tsuruta et al. [19] found genetic correlations of −0.28, −0.34 and −0.19 by a bivariate threshold model analyzed by a Gibbs sampling algorithm. These values are clearly much lower than the results of the current study. The genetic correlation with somatic cell score was −0.11—that is, higher LIV, less udder disease. Female fertility was positively correlated with LIV, and persistency of milk production was negatively correlated. The genetic correlations with the calving traits were nearly zero. In the sire evaluations, dystocia was negatively correlated with LIV, as less dystocia should result in less cow death. For the milk production traits, persistency and herd life, the correlations with the sire breeding values derived from data set 6 were in the same directions as the genetic correlations. 

The contribution of each of the index traits to the proposed index was computed as described in equation (4) using the values listed in Table 2. As shown in Table 2, LIV would account for 9% of the proposed index. This value can be compared to the US indexes, in which LIV was allotted only 1% of the total [8]. The expected genetic gains after 10 years of selection with the current index, in which LIV has a coefficient of zero, and the proposed index with a coefficient of 48, are given in Table 2. The expected change for LIV of −2.33% over 10 years can be compared to the annual genetic trend of −0.22% per year, that is, −2.2% in 10 years, derived from data set 5. This discrepancy of 0.01%/year is insignificant compared to the genetic standard deviation of the trait, 2.93. Implementation of the proposed index would result in a reduction of 1.2% in the decline in LIV over 10 years, but significantly lower genetic gains for the milk production traits, herd life and milk production persistency. The changes in the calving traits were in the economically favorable direction, but were economically insignificant. Using a coefficient of 5 for LIV, the expected change for this trait would be 0.1%. Changes in the other traits were also insignificant.

### 3.3. Genomic Analysis of Livability

The Manhattan plot for the GWAS results for LIV is given in Figure 5. By genomic REML analysis, we found that 70% of the variance among the sires’ transmitting ability could be explained by considering all SNPs. While accounting for relationships and after accounting for multiple tests, we found seven significant markers with adjusted *p*-values ≤ 0.05 (Table 5). The SNP with the lowest probability of accepting the null hypothesis, ARS-BFGL-NGS-363 (*p* = 1.82 × 10^−42^), is located on chromosome 13, in close proximity to the bovine gonadotropin-releasing hormone 2 (*GNRH2*), oxytocin/neurophysin I prepropeptide (*OXT*) and arginine vasopressin (*AVP*) genes. *OXT* and *AVP* are important to smooth muscle contraction during parturition and lactation and are associated with endometritis, as summarized in GeneCards [31].

We performed QTL and gene enrichment analyses to annotate the significant markers’ possible roles in LIV. The confidence interval for each significant marker harbors several genes and QTL that have been previously reported. We thus analyzed the distribution of the genomic distances between markers as a function of the R^2^ values for genetic linkage. About 85% of the marker pairs with R^2^ > 0.9 are within 150 kb distance from each other, as we found recently in this population [32]. We therefore obtained QTL and genes within an interval of 150 kb spanning the significant markers for enrichment analysis. QTL enrichment analysis found significant LIV markers are mainly enriched with fat and protein yield, and content QTL, in concordance with the genetic correlations given in Table 4, but also with QTL for tick resistance, body weight, fertility, respiratory disease susceptibility and residual feed intake (Figure 6).

Gene enrichment analysis of the genes spanning the significant markers for LIV revealed significant enrichment to the vasopressin and the opioid prodynorphin pathways, and for the GO interleukin-17 (IL-17) signaling process (Table 6). IL-17s are important cytokines for protecting against infection and intestinal mucosal inflammation [33]. In addition, it has been shown previously that disruption of the bovine *OXT* and vasopressin pathways results in severe pathology at parturition that can finally lead to maternal death [34]. This finding is in accordance with the results from the phenotypic analysis, in which cow MORT was highest during the first month after calving.

## 4. Discussion

As noted in the introduction, it was necessary to determine the effects of environmental factors on LIV, in order to compute unbiased evaluations for LIV, which were required for the GWAS analysis. In general, nongenetic effects on LIV, including HYS, culling month, months in milk at culling, final parity, pregnancy status at culling and milk production in the final lactation were similar to those in previous studies [6,7,9,10]. VanRaden et al. [8] found in the US Holstein population that for bulls recently in general service with >80% reliability, genetic evaluations for LIV were correlated at about 0.70 with productive life, 0.45 with the daughter pregnancy rate and −0.25 with somatic cell score, but correlations were low for milk production traits. These values are clearly different from the current results. The major effect of pregnancy on LIV has not been reported previously, and the large negative correlations with the milk production traits found in this study are unique.

According to selection index theory, if selection is based on genetic evaluations, maximum economic progress is obtained when the selection index coefficients correspond to the economic values of the traits included in the index. Thus, a selection index that includes the correct economic value for LIV should result in greater total economic gain as compared to an index with a coefficient of zero. Based on the value of $1200 per cow death, a gain of 1.3% in LIV has an economic value of $16/cow. It is unlikely that milk producers would be willing to forgo an 11 kg increase in protein production to achieve the predicted LIV gain. It should be noted, though, that the prediction given in Table 2 is based on first parity phenotypic records for all traits, except for LIV and herd life. With an index coefficient of five, LIV accounts only for 1% of the index, as proposed by [8], but in this case, changes in expected gains are insignificant. 

A total of seven SNPs passed the threshold of adjusted *p*-values ≤ 0.05 (Table 5) in the GWAS, with substitution effects ranging 0.68 to 1.76. All the SNPs markers included in the GWAS analysis explained jointly 70% of the variation in the bull’s transmitting ability. Tsuruta et al. [10] and Freebern et al. [35] computed GWAS analyses for LIV in the US Holstein population. The only significant effect found by [10] was on chromosome 14, adjacent to the *DGAT1* gene (diacylglycerol O-acyltransferase 1), which is known to affect fat concentration. Freebern et al. [35] found six regions with significant effects for cow LIV after correction for multiple comparisons on chromosomes 5, 6, 14, 18, 21 and 23. None of these effects correspond to the effects found in the current study.

The functional analysis of the significant markers pointed to the IL-17 signaling and the vasopressin pathways. The importance and implications of the IL-17 cytokines in health and disease were demonstrated in cattle. For instance, activation of IL-17 is required upon infection with paratuberculosis and Johne’s disease, in Salmonella-induced gut inflammation and in pathogens-derived mastitis [36]. In addition, the activity of IL-17 in animals during infection was shown to be essential in the modulation of viral infection and pathogenesis of the respiratory system [37]. This is in concordance with our QTL enrichment analysis that found significant enrichment of significant SNPs with bovine respiratory disease susceptibility QTLs. These suggest a possible role of the IL-17-mediated immune response in MORT. The ARS-BFGL-NGS-363 marker in the bovine chromosome 13 has the highest substitution effect (1.76, *p* = 1.82 × 10^−42^) and is located in close proximity to the genes *OXT* and *AVP*, which are both part of the vasopressin pathway. The bovine *OXT* plays an essential role in maternal lactation, but apparently also in parturition. The disruption of the bovine *OXT* in the transgenic mouse model resulted in severe pathology at parturition [34], suggesting a possible role in the increased MORT at early lactation which was observed. In addition to the function of *OXT* in parturition, previously, *AVP* and *OXT* were shown to each have a role in regulating postpartum estrous behavior in dairy cows, and possibly to be involved in the calving interval and timing of the first insemination postpartum [38]. In accordance, genetic factors that shorten the calving or conception intervals may affect MORT. This is in line with the differences in the LIV-dependent pregnancy status we found.

## 5. Conclusions

Even though LIV has significant economic value, inclusion in the Israeli breeding index is not recommended. A marker in proximity to the oxytocin-vasopressin locus had the greatest effect in the GWAS analysis. Previous studies show that oxytocin activity in cattle affects calving-associated pathologies and maternal death. It may be possible to determine the causative mutation by sequencing of individual bulls for the chromosomal segment of interest.

## Figures and Tables

**Figure 1 genes-14-00588-f001:**
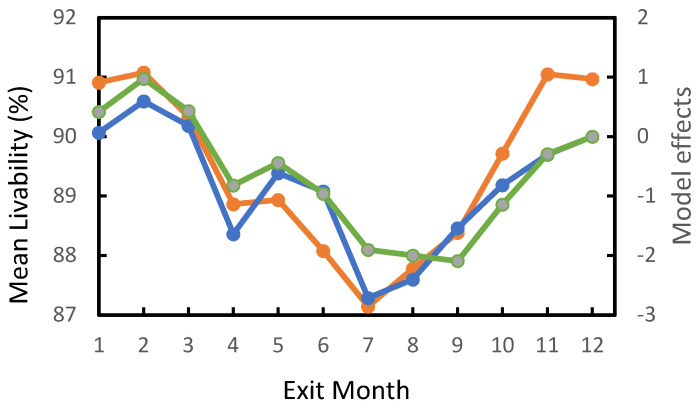
Mean LIV scores and model effects by exit month. Orange line, mean LIV; green line, GLM model effects from the analysis of data set 1; blue line, model effects from the REML analysis of data set 3.

**Figure 2 genes-14-00588-f002:**
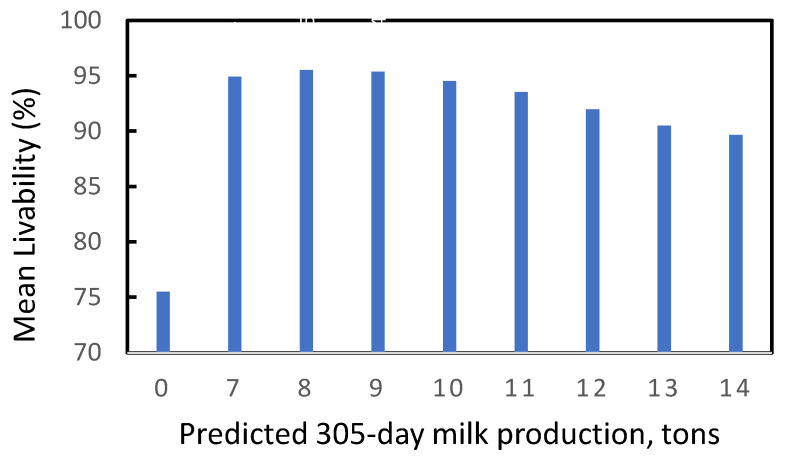
Mean LIV score as a function of predicted 305-day milk production in the final lactation. The column for zero production represents cows that were culled or died prior to computation of a milk production record.

**Figure 3 genes-14-00588-f003:**
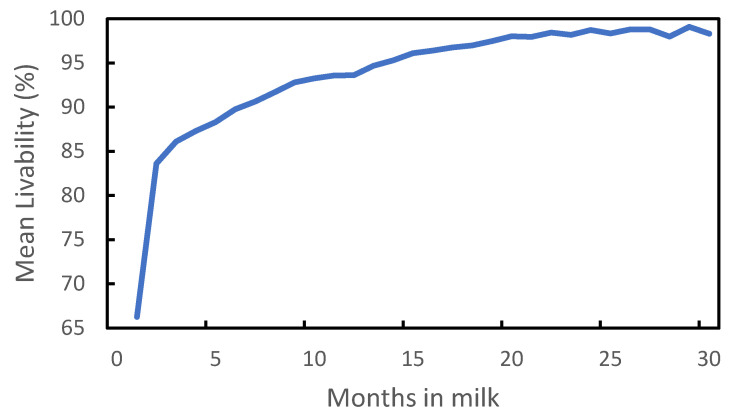
Mean livability as a function of months in milk.

**Figure 4 genes-14-00588-f004:**
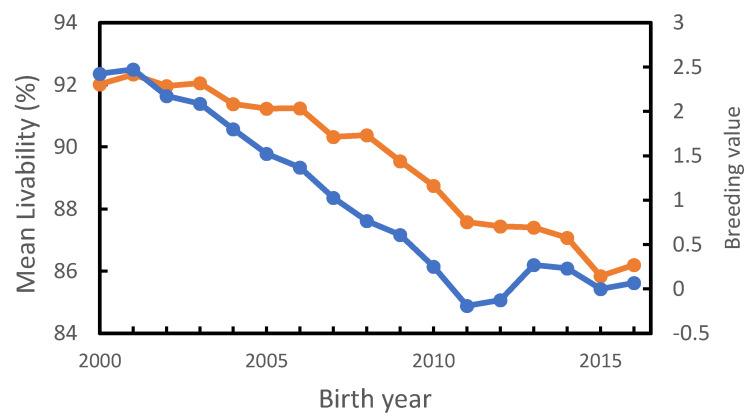
Mean livability scores of cows included in data set 5 by birth year, orange line; and mean their breeding values, blue line.

**Figure 5 genes-14-00588-f005:**
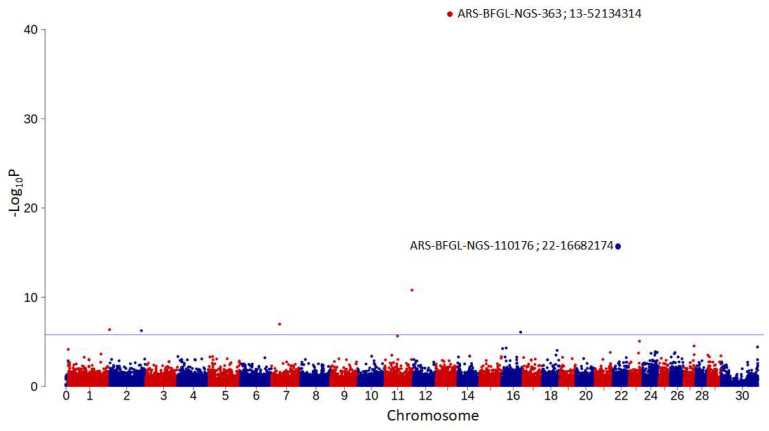
Manhattan plot for livability. The markers’ chromosomal positions, annotated on the ARS-UCD1.2 genome assembly, are on the x-axis, and the nominal −log_10_ *p*-values are on the y-axis. Chromosome 0 denotes markers with unknown map positions, and chromosome 30 is the sex chromosome. The blue horizontal line indicates the genome-wide significance threshold of *p* = 0.05 after correction for multiple testing. The marker labels and map positions for the two markers with the lowest nominal *p*-values are listed.

**Figure 6 genes-14-00588-f006:**
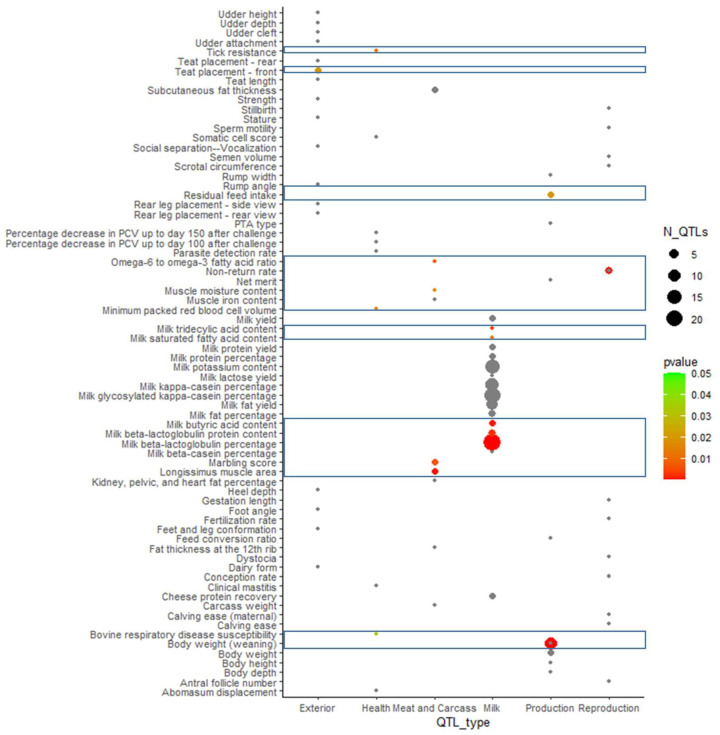
Bubble plot presenting the enrichment of previously reported QTL with the identified LIV significant markers. The color denotes the enrichment-adjusted *p*-value score, and the area of the circles is proportional to the number of QTL (N_QTL). The y-axis represents the quantitative traits, and the x-axis denotes the affiliation of the QTL with the major trait class. Blue boxes mark clusters of significant QTL.

**Table 1 genes-14-00588-t001:** The basic statistics of the 7 data sets analyzed.

	Animals	Analysis Type	Numbers of:			
Data Set	Analyzed		Records	Ancestors	Herd-Year-Seasons	Genetic Groups
1	Cows	Linear model	523,954	-	-	-
2	Cows	Pearson correlation	475,512	-	-	-
3	Cows and bulls	REML ^1^	174,106	123,596	23,695	2
4	Cows and bulls	REML	145,023	116,141	9507	2
5	Cows and bulls	IAM ^2^	511,438	171,722	49,425	64
6	Bulls	Pearson correlation	1007	-	-	-
7	Bulls	GWAS ^3^	965	-	-	-

^1^ Restricted maximum likelihood analysis by the individual animal model. ^2^ Individual animal model analysis. ^3^ Genome-wide association study.

**Table 2 genes-14-00588-t002:** Comparison of current breeding index, PD20, and the proposed index, including livability.

Trait	Index	Genetic	Fraction of Index		Expected Changes after 10 Years	
	Coefficient ^1^	SD	PD20	Proposed Index	PD20	Proposed Index
Milk (kgs)	0	999	0.000	0.000	1197	827
Fat (kgs)	9.94	37	0.256	0.233	53	39
Protein (kgs)	19.88	25	0.352	0.320	38	27
Somatic cell score ^2^	−300	0.50	0.106	0.096	−0.14	−0.13
Fertility (%)	26	7.88	0.145	0.131	0.45	0.46
Herd life (days)	0.6	174	0.074	0.067	137	106
Persistency (%)	10	5.57	0.039	0.036	2.04	1.22
Maternal dystocia (%) ^2^	−3	5.99	0.013	0.012	−0.96	−0.77
Maternal stillbirth (%) ^2^	−6	3.75	0.016	0.014	−0.61	−0.48
LIV (%)	0.48	2.93	0.000	0.090	−2.33	−1.05

^1^ The index coefficient for livability is zero in the current index and 48 in the proposed index. ^2^ For these traits, negative values are economically favorable.

**Table 3 genes-14-00588-t003:** Simple means and frequencies from data set 1 and model effects for pregnancy status and parity for the analyses of data sets 3 and 5.

		Data Set 1		Effects ^1^	
Effect	Level	Percent of Cows ^2^	Means	Data Set 3	Data Set 5
Pregnancy status	open	89.79	91.1	14.1	11.9
	pregnant	10.21	76.3	0	0
Exit parity	1	25.99	91.0	3.3	5.1
	2	23.40	91.5	5.0	5.8
	3	19.84	89.6	3.5	3.8
	4	15.29	87.5	1.7	1.7
	5	9.95	86.6	0.9	0.7
	6	5.52	86.3	0	0

^1^ Effects are computed relative to the last level. ^2^ Refers to status at death or exit.

**Table 4 genes-14-00588-t004:** Phenotypic, environmental and genetic correlations of livability with the nine index traits.

Trait	Data Set 2	Data Set 4		Data Set 6
	Phenotypic	Environmental	Genetic	Sire Evaluations
Milk	−0.07	−0.06	−0.46	−0.46
Fat	−0.06	−0.04	−0.32	−0.27
Protein	−0.07	−0.05	−0.42	−0.48
Somatic cell score *	0.01	0.01	−0.11	0.07
Female fertility	−0.04	−0.02	0.04	−0.04
Herd life	−0.02	0.05	−0.06	−0.23
Persistency	−0.02	−0.02	−0.26	−0.28
Maternal dystocia *	0.01	0.00	0.00	−0.09
Maternal stillbirth *	0.00	0.00	0.01	−0.01

* Negative values are economically favorable.

**Table 5 genes-14-00588-t005:** Single nucleotide polymorphisms associated with livability with adjusted *p*-value ≤ 0.05.

SNP ^1^	CHR ^2^	POS ^2^	BETA ^3^	*p*-Value ^4^
ARS-BFGL-NGS-363	13	52,134,314	1.76	1.82 × 10^−42^
ARS-BFGL-NGS-110176	22	16,682,174	1.17	1.77 × 10^−16^
ARS-BFGL-NGS-15093	11	10,298,952	0.91	1.61 × 10^−11^
ARS-BFGL-NGS-36142	7	30,866,207	1.05	1.03 × 10^−7^
ARS-BFGL-NGS-105119	1	157,725,088	0.77	4.20 × 10^−7^
ARS-BFGL-NGS-114262	2	120,130,818	0.68	5.54 × 10^−7^
ARS-BFGL-NGS-15423	16	72,258,249	0.72	8.12 × 10^−7^

^1^ Markers are sorted in descending order of the nominal *p*-value. ^2^ Marker coordinate in base pairs according to the bovine ARS-UCD1.2 assembly. ^3^ Substitution effect in units of the sires’ transmitting ability for livability in percent. ^4^ The EMMAX GWAS nominal *p*-value.

**Table 6 genes-14-00588-t006:** Genes enrichment analysis. Top-scored Panther pathways and Gene Ontology (GO) processes obtained by the Enrichr algorithm.

Name ^1^	Adjusted *p*-Value ^2^	Score ^3^
Vasopressin synthesis Homo sapiens P04395	0.002	852
Opioid prodynorphin pathway Homo sapiens P05916	0.004	363
p53 pathway by glucose deprivation Homo sapiens P04397	0.094	60
Nicotinic acetylcholine receptor signaling pathway Homo sapiens P00044	0.038	54
Hypoxia response via HIF activation Homo sapiens P00030	NS ^4^	50
interleukin-17-mediated signaling pathway (GO:0097400)	0.042	1000
cellular response to interleukin-17 (GO:0097398)	0.042	1000
nucleoside triphosphate metabolic process (GO:0009141)	NS	520
protein polyufmylation (GO:1990564)	NS	446
positive regulation of fertilization (GO:1905516)	NS	446

^1^ Name of the Enrichr Panther pathway or GO process with accession number in parentheses. ^2^ Enrichr adjusted *p*-value as derived from the Fisher exact test after correction for multiple testing. ^3^ The Enrichr combined score as derived from the Fisher exact test Log(p)*Z-score, where the Z-score is the deviation produced from the empirical distribution of the Fisher exact test from random input gene lists on the same term. ^4^ Not significant.

## Data Availability

Restrictions apply to the availability of these data. The data were obtained from the database of the Israel Cattle Breeders Association (ICBA) and are available from the authors with the permission of ICBA.

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
