# Peer review of "Genetic and Genomic Analysis of Cow Mortality in the Israeli Holstein Population"

_genes, 2023, doi:10.3390/genes14030588_

Round 1

Reviewer 1 Report

This is an interesting and well-written manuscript aimed to analyze genetic and genomics of cow mortality in the Israeli Holstein population. The methodology of the study included several strong datasets. However, the main concern is that of the 7 objectives described in this study, only the last one involves genes or genomics associated with the studied trait (i.e., livability). I suggest the authors to describe how the 6 initial objectives, which are related to phenotypic and genetic analyses, contribute to or support the final objective. This would help to establish a connection of the whole study with the genomic analysis performed and with the scope of the Journal.

I suggest considering next major issues:

1)    Abstract: According to guidelines, this section should describe briefly the main methods used in the study and to include a final conclusion, which are missing.

2)    Materials and methods: For GWAS study, I suggest to provide answer for next questions:

-       How the DNA was extracted?

-       How the DNA quality was confirmed?

-       What were the criteria to ensure quality of SNP selected for GWAS?

-       What was the genome-wide significance threshold (-Log10P)?

-       What was the reference for the “Enrich server” used for the gene enrichment analyses?

3)    Results: Please correct the number of the sub-headings as they must be progressive.

4)    Discussion: In this section, the connection between phenotypic/genetic analyses with the genomic study is not clear. Please clarify how the phenotypic and genetic studies contributed with the scope of the journal.

I suggest improving discussion about SNP or genes associated with livability.

5)    Conclusions: This section appears to be a brief summary of the results. Instead, I suggest including 2 or 3 conclusive sentences describing the relevance of the study or the importance of the findings.

6)    References: In several references is necessary to correct Journal name and Journal year following the guidelines described in “Instructions for authors”.

Reviewer 2 Report

Please, correct the abstract and the conclusions. They need to differ in the content and meaning. 

Author Response

Reply to the reviewer's comments are in the attached file.

Round 2

Reviewer 1 Report

The manuscript has improved significantly. However, there are some minor issues that should be considered:

- A brief description of the methods is still missing in the Abstract.

- I suggest to consider next questions in Materials and Methods: How blood samples were collected, where DNA extraction was performed? 

- Font size of the reference 17 (line 551) should be adjusted.
